# Rotational Molding of Poly(Lactic Acid)/Polyethylene Blends: Effects of the Mixing Strategy on the Physical and Mechanical Properties

**DOI:** 10.3390/polym13020217

**Published:** 2021-01-09

**Authors:** Eduardo Ruiz-Silva, Mirleth Rodríguez-Ortega, Luis Carlos Rosales-Rivera, Francisco Javier Moscoso-Sánchez, Denis Rodrigue, Rubén González-Núñez

**Affiliations:** 1Departamento de Ingeniería Química, Universidad de Guadalajara, Blvd. Gral. Marcelino García Barragán #1451, Guadalajara, Jalisco 44430, Mexico; lalo.ruiz1989@gmail.com (E.R.-S.); mirlethrodriguez@gmail.com (M.R.-O.); 2Department of Chemical Engineering and CERMA, Université Laval, Quebec, QC G1V 0A6, Canada; denis.rodrigue@gch.ulaval.ca

**Keywords:** rotational molding, poly(lactic acid), thermal degradation, mechanical properties, mixing strategies

## Abstract

In this study, blends of poly(lactic acid) (PLA)/linear medium density polyethylene (LMDPE) at different weight ratios were prepared by rotational molding. Two mixing strategies were used to evaluate the effect of phase dispersion on the physical and mechanical properties: (i) Dry-blending (DB) using a high shear mixer, and (ii) melt-blending (MB) using a twin-screw extruder. Thermal, morphological, and mechanical analyses were performed on the neat polymers and their blends. The thermal analysis was completed by differential scanning calorimetry (DSC) and thermogravimetric analysis (TGA), and the blends prepared by MB had lower thermal stability than the ones prepared via DB due to some thermo-oxidative degradation through the double thermal process (extrusion and rotomolding). The morphology of the rotomolded parts showed that DB generated larger particle sizes (around 500 µm) compared to MB (around 5 µm) due to the shear and elongational stresses applied during extrusion. The tensile and flexural properties of the rotomolded parts combined the PLA stiffness with the LMDPE toughness independent of the blending technique. Neat PLA presented increments in tensile strength (54%) and flexural strength (111%) for DB compared with MB. A synergistic effect in impact strength was observed in blends with 12 and 25 wt. % of PLA prepared by DB.

## 1. Introduction

Rotational molding is a process to manufacture hollow plastic parts over a wide range of sizes, shapes, and thicknesses [1]. This method presents important advantages over other technologies such as parts without weld lines having low residual stresses combined with lower capital investment costs and greater flexibility for colors and materials changes from part to part [2]. The global market for rotational molding is around 1000 ktons per year, where the U.S. represents half of the world’s consumption [3]. Nevertheless, the potential of rotational molding as a manufacturing process for plastic products is limited by some aspects such as difficulties in process control, materials in powder form, long cycle times, and a narrow range of polymers. In fact, most resins used in rotational molding are thermoplastic powders, of which the different grades of polyethylene (PE) represent more than 85% [4]. Some other materials available for rotational molding include polyvinyl chloride, polypropylene, acetate butyrate, polycarbonate, and polyesters, among others [5]. Poly(lactic acid) (PLA) is considered a promising alternative to petrochemical-derived polymers in a wide range of commodity and engineering applications [5]. Its production capacity is expected to be over 800,000 tons by 2020 [6].

PLA can be produced by ring-opening polymerization of lactide or condensation polymerization of lactic acid monomers, both performed via fermentation of renewable resources [7]. Melt processing is the main technique used for mass production of PLA products for the medical, textile, plasti-culture, and packaging industries [7,8]. The main advantages of PLA are its biodegradability, transparency, and good mechanical properties that are comparable to those of polystyrene (PS) and poly(ethylene terephthalate) (PET) [9,10]. However, some disadvantages, such as low thermal stability, high moisture sensitivity, low impact strength, and brittleness, have limited its applications [11,12]. To overcome these drawbacks, natural fibers addition [13] and blending with other polymers [14], such as thermoplastic starch [15] and polyolefins [16,17], have been proposed.

Physical blending is a convenient route to develop new polymer materials by combining the properties of each polymer. This method is usually cheaper and takes less time than the development of new monomers and/or new polymerization routes. Furthermore, the resulting blend properties can be adjusted through the suitable selection of components and composition, therefore obtaining a final product with desired characteristics like mechanical, morphological, and thermal properties [10]. Recently, Zhao et al. [5] presents a review of super tough PLA blends that summarize and organize the current development in super tough PLA fabricated via polymer blending. For example, Zeng et al. [18] prepared PLA/EVM (ethylene-co-vinyl acetate) blends in a mixing chamber and followed with thermocompression. The toughness of PLA was improved by incorporation of rubber grade ethylene-co-vinyl acetate (EVM). PLA/Polyolefin blends have been developed [19,20]; for example, Anderson and Hillmyer [21] studied the influence of Polyethylene on the structure and mechanical properties of PLA/polyethylene blends. A diblock copolymer (PE-*b*-PLLA) was used as compatibilizer, and toughening was achieved without a copolymer, and the variability of impact resistance decreased with as little as 0.5% of copolymer. Kim et al. [22] studied PLA/LDPE blends in order to improve the brittleness of PLA, and they used a grafted copolymer of PE-g-glycidyl methacrylate (PE-GMA). Their results show that PE-GMA reduces the domain size of dispersed phase and enhances the tensile properties of PLA/LDPE blends. 

PLA has been processed by almost all melt processing techniques [7]. The process consists of heating the polymer above its melting temperature, molding it into the desired shape, and cooling it, and this implies a control in the thermal properties, crystallization, and rheological properties. Lim et al. [23] presents an interesting review of processing technology for PLA. This review includes a discussion on the preparation of PLA materials, for example extruded cast and oriented films, and melt-spun fibers for nonwovens, textiles, and carpets, injection molded disposable cutlery, injection stretch blown bottles, and thermoformed containers and cups [23]. 

According to our knowledge, there is very limited information about PLA-based polymer blends being processed via rotomolding. Greco and Maffezzoli [24,25,26] showed that through the proper combination of material viscosity and processing parameters, it was possible to use PLA to manufacture biodegradable containers via rotational molding. Recently, Cisneros-López et al. [27] produced rotomolded parts of neat PLA and PLA/agave fiber biocomposites by dry-blending followed by rotational molding. Their results showed that these biocomposites have high porosity and low properties due to poor adhesion between PLA and agave fibers.

The aim of this work is to demonstrate an efficient mixing approach for the preparation of PLA/LMDPE rotomolding to ensure a more uniform and better dispersed phase in the blends and toughen the PLA. The methods used were: (i) Dry-blending (DB) using a high shear mixer and (ii) melt-blending (MB) using a twin-screw extruder. The blends were then processed by rotational molding using a laboratory scale biaxial machine and followed via internal air temperature. Finally, characterization in terms of morphology (SEM and optical microscopy), thermal (DSC and TGA), and mechanical (tensile, flexural, and impact) properties is performed.

## 2. Materials and Methods 

### 2.1. Materials

Rotational molding grade LMDPE RO93650 in powder form was supplied by Polímeros Nacionales (Guadalajara, Jal, Mexico) and used as received. This polymer has a melt flow index (MFI) of 5.0 g/10 min (190 °C, 2.16 kg) and a density of 0.936 g/cm^3^, according to the provided technical data sheet. PLA Ingeo 3251D from Nature Works LLC (Blair, NE, USA) with a MFI of 35.0 g/10 min (190 °C, 2.16 kg) and a density of 1.24 g/cm^3^, according to the provided technical data sheet and supplied as spherical pellets. As a demolding agent, silicone spray lubricant 3A-RP was supplied by Poliformas Plásticas (Guadalajara, Jal, Mexico) and used to prevent the parts sticking to the mold surfaces.

The PLA was firstly pulverized with an ultra-centrifugal mill Retsch ZM 200 (Haan, Germany) in which the size reduction takes place by impact and shearing effects between the rotor and the fixed ring sieve. Furthermore, to facilitate the grinding process and avoid excessive wear of the cutting elements, the PLA was placed in an Arctiko ultra-low temperature freezer (Esbjerg, Denmark) at a temperature of −80 °C at least 24 h before grinding. Table 1 shows the particle size distribution of the used powder polymers measured using a standard (ASTM B214) sieve analysis.

### 2.2. Blend Preparation

PLA/LMDPE blends with different weight ratios were prepared by dry-blending (DB) using a high shear mixer JR Torrey LP-12 (Monterrey, NL, México) at 3750 rpm for 5 min. For melt-blending (MB), a co-rotating twin-screw extruder Thermo Fisher Scientific (Karlsruhe, Germany) Process 11 with an L/D of 40, 2 mm diameter circular die, and 8 electrical heated zones was used. The temperature profile from the feed hopper to the die was 160/160/170/170/180/180/190/190 °C with a screw rotational speed of 60 rpm. The extruded material was cooled in a water bath and then pelletized. Subsequently, the pellets were pulverized by the same grinding process as described for neat PLA. All the polymers were oven dried at 60 °C for at least 24 h before carrying out rotational molding. Sample codes with their composition are reported in Table 2.

Rotational molding was carried out in a laboratory-scale machine with a stainless-steel mold in the shape of a cubic prism of 2 mm wall thickness, 15 cm per side at the top and bottom, and 16 cm per side in the central area. The demolding agent was sprayed to the internal surface of the mold before loading the material (350 g in all cases). Then, the loaded mold was closed, mounted on the rotating arm, and introduced into the oven which was previously heated at 300 °C. The mold was kept rotating for 30 min at 300 °C (heating cycle). The internal air temperature (IAT) was monitored with a thermocouple through the vent during the whole process. After the heating cycle, the mold was removed from the oven and cooled down by forced air convection until the IAT dropped to 60 °C. Finally, the mold was opened, and the part was demolded. During the rotomolding process (heating and cooling), a rotational speed for the major axis (arm speed) and the minor axis (plate speed) was set to 2.70 and 3.5 rpm, respectively, according to López-Bañuelos et al. [2].

### 2.3. Rheological Characterization

Dynamic oscillatory shear measurements of the neat rotomolded polymers were performed at 180 °C, 200 °C, and 220 °C using an ARES rheometer from TA Instruments (New Castle, DE, USA). A parallel plate geometry of 25 mm in diameter and 1.5 mm gap was used. Strain sweep tests were performed first to confirm that the experimental data were taken in the linear viscoelastic zone of the material. Then, frequency sweep tests between 0.04 and 300 rad/s were performed using a 5% deformation.

### 2.4. Morphology

Micrographs of the rotomolded parts prepared by DB were taken using an optical stereomicroscope Olympus SZ6 coupled with a Spot Insight (Tokyo, Japan) high resolution (1600 × 1200 active pixels) camera. On the other hand, the blends prepared by MB were analyzed on a field emission scanning electron microscope (FE-SEM) TESCAN MIRA3 LMU (Warrendale, PA, USA) using a laser beam of 10 kV accelerating voltage. Prior to SEM observation, the samples were immersed in liquid nitrogen, fractured, and finally coated with a thin conductive gold layer under vacuum using a SPI Module Sputter Coater for 60 s to prevent surface charging. The particle sizes of the dispersed phase of the samples prepared by DB and MB and processed by rotomolding were measured using Image-Pro Plus 4.5 (Media Cybernetics, Rockville, MD, USA).

### 2.5. Density

Density was measured by a gas pycnometer ULTRA-PYC 1200e, Quantachrome Instruments (Boynton Beach, FL, USA) using nitrogen. The data reported are the average of three measurements.

### 2.6. Thermal Properties

#### 2.6.1. Differential Scanning Calorimetry (DSC)

A differential scanning calorimeter TA Instruments Discovery (New Castle, DE, USA) was used to study the thermal properties of the blends. The samples weight was approximately 10 mg. All the samples were taken from the rotomolded parts and placed in aluminum pans for solids (TA Instruments T160606, USA). The specimens were heated at 10 °C/min from 25 to 200 °C under a nitrogen flow of 50 mL/min. Glass transition temperature $\left(T_{g}\right)$, cold crystallization temperature $\left(T_{cc}\right)$, melting temperature $\left(T_{m}\right)$, enthalpy of cold crystallization $\left(\Delta H_{cc}\right),$ and melting enthalpy $\left(\Delta H_{m}\right)$ were determined from the DSC curves. The values for $T_{m}$ and $\Delta H_{m}$ were taken as the peak temperature and the area of the melting endotherm, respectively.

Crystallinity levels $(X_{c})\;$ of the samples were evaluated from their corresponding melting enthalpies as [23]:(1)$$X_{c}\left(\%\right)=\frac{\Delta H_{m}-\Delta H_{cc}\;}{\Delta H{}^{\circ}}\times \frac{100}{w}$$
where ΔH°(PLA) and ΔH°(LMDPE) are the enthalpies of melting per gram of 100% crystalline (perfect crystal) PLA and LMDPE (93.7 [28] and 288 J/g [16]), respectively, while w is the weight fraction of the component in question in the blend.

#### 2.6.2. Thermogravimetric Analysis (TGA)

The decomposition behavior and thermal stability of the rotomolded parts were analyzed using a thermogravimetric analyzer model Q5000 IR from TA Instruments (New Castle, DE, USA). Samples of approximately 20 mg were heated at 10 °C/min from 50 to 600 °C under an air flow of 50 mL/min.

### 2.7. Mechanical Properties

All the specimens for the mechanical tests were cut from the rotomolded parts with a laser machine Guian ((Dongguan, China) GN-640MS and tested at room temperature. Tensile and flexural properties were evaluated using an Instron 3345 universal testing machine (Norwood, MA, USA). Tensile tests were carried out on dog bone samples according to ASTM D638 (type IV) at a crosshead speed of 5 mm/min. Reported values for modulus and strength are based on the average of at least five samples. Flexural tests were performed according to ASTM D790 with a crosshead speed of 2 mm/min. Samples dimensions were 70 × 12.7 × depth mm^3^ and the span length was fixed at 16 times the sample depth. At least five samples were used to report the average and standard deviation for modulus and strength. Charpy impact strength was determined by a CEAST 9050 impact tester from Instron (Norwood, MA, USA) according to ASTM D6110. A CEAST 6897 manual sample notcher from Instron (Norwood, MA, USA) was used to notch the samples at least 24 h before testing. Each value represents the average of 8 notched samples. In order to better understand the effect of PLA or LMDPE on the mechanical properties, two additional compositions closer to the neat materials were included to obtain more information on the effect of the mixing strategy on the performance of rotomolded materials PLA87 and PLA12.

## 3. Results and Discussion

### 3.1. Rheological Properties

Figure 1 shows the complex viscosity curves as a function of frequency for (Figure 1a) neat PLA and neat LMDPE at 180 °C, 200 °C, and 220 °C and in Figure b, the blends are presented at 200 °C. The effect of temperature on Newtonian viscosity ($\eta_{0}$) was related to an Arrhenius equation as:(2)$$\eta_{0}\left(T\right)=\eta_{0}\left(T_{0}\right)exp\left[\frac{E_{a}}{R}\left(\frac{1}{T}-\frac{1}{T_{0}}\right)\right]$$
where R is the universal gas constant. The activation energy $\left(E_{a}\right)$ was calculated using 200 °C as the reference temperature $\left(T_{0}\right)$. Slightly lower values were obtained for DB compared to MB 105.5 and 115.6 kJ/mol for PLA, and 24.5 and 25.1 kJ/mol for LMDPE.

Figure 1a shows that PLA has lower viscosity than LMDPE over the frequency range. Moreover, increasing temperature has a greater effect on PLA viscosity. This may turn out to be unsuitable for the manufacture of parts with uniform thickness in rotational molding where the materials undergo high processing temperatures [4]. On the other hand, at 220 °C, the viscosity of the PLA prepared by MB was found to be lower than the viscosity of the PLA prepared by DB. This can be attributed to thermo-mechano-oxidative degradation due to both processing steps (extrusion and compression molding) imposed on the material, and similar results have been reported [29,30,31], a lower PLA viscosity is related to lower molecular weight after processing. As expected, LMDPE viscosity showed less dependence on the mixing method mainly attributed to a higher thermal stability [32]. The results for the blend complex viscosity shown in Figure 1b depicts a clear overall decrease in the viscosity of the mixtures prepared by MB, which can also be attributed to a double melt processing. In addition, 25% PLA blend has a Newtonian viscosity slightly higher than that of LMDPE. Zhou et al. [33] reported a synergistic effect on the rheological properties of PLA/LDPE blends, which were attributed to the entanglements of long branched molecular chains in LDPE and linear molecular chains in PLA.

### 3.2. Internal Air Temperature Profiles

Rotational molding of semi-crystalline polymers can be divided into six different stages, each one being related to a change in the internal air temperature (IAT) slope [4,34,35]. The IAT variation is proportional to thermal changes occurring in the polymer (melting-sintering, crystallization and solidification, etc.), which are possible to follow and control/optimize the complete rotomolding cycle.

Recently, González-Núñez et al. [35], using the first and second derivatives, analyzed the IAT profiles of un-foamed and foamed PE parts obtained by rotational molding. The use of this tool allowed them to accurately obtain relevant information about the different transitions occurring in the polymer throughout the rotomolding cycle. Figure 2a presents the IAT profiles for PLA, LMDPE, and the blends prepared by DB. In addition, the first (Figure 2b) and second (Figure 2c) derivatives of these profiles are also reported. The IAT profiles and their derivatives for blends prepared by MB present similar trends, but for simplicity are not shown.

The neat PLA has an initial increase in the slope of the first derivative at 10 min (around 105 °C) labeled as point A related with powder adhesion to the mold wall. As the temperature inside the mold increases, another inflection point is observed (Point B) associated with the melting point of the polymers. For LMDPE, the inflection point is located at 14.7 min (122 °C), while it is shifted to 15.5 min (160 °C) for PLA. Once all the material is melted, the densification stage takes place, and during this stage the material is consolidated and the IAT continues to increase until the peak internal air temperature (PIAT, point C) is reached. This point occurs slightly after the mold is removed from the oven and the cooling stage begins. Between points D and E, a change in the cooling rate related to the crystallization process occurs. After the crystallization/ solidification step, the IAT continues to decrease until the demolding temperature (60 °C) at point F [1,25,36,37].

In Figure 2a, it is possible to observe how the IAT profiles change as the PLA content in the blend increases. The PIAT increased from 225 °C for the LMDPE to 257 °C for the PLA. This increase can be related to the differences in the thermal properties of the neat polymers (thermal diffusivity). The melting enthalpy of PLA is also lower than that of the LMDPE (41.8 and 130.8 J/g, respectively, from DSC results), requiring a smaller amount of the heat supplied for PLA melting compared to the heat required for LMDPE melting. The experimental measured density of the polymers is another factor affecting the IAT profiles. Since PLA has a higher density than LMDPE, this decreases the average part thickness as the amount of PLA in the blend increases (see Table 3) independently of the mixing strategy. In the heating and cooling stages, the lower thickness enables a higher heat transfer to/from the mold, increasing the PIAT and decreasing the total cycle time. As mentioned above, the lower PLA viscosity when processed at higher temperatures results in a poor distribution in the rotomolded parts. Therefore, the use of PLA/LMDPE blends produced parts with a more uniform thickness due to the better processability of LMDPE. In the case of densities, the values for the different mixing methods are within the experimental error (Table 3).

### 3.3. Differential Scanning Calorimetry

The DSC thermograms for PLA, LMDPE, and their blends prepared by both mixing methods are shown in Figure 3. The observed values of the glass transition, transition temperatures, the corresponding enthalpy values, and crystallinity level $\left(X_{c}\right)$ for each composition are summarized in Table 4. In all cases, the blends present a phase separation identified by two endotherms around 128 and 167 °C, corresponding to the melting temperatures $\left(T_{m}\right)$ of the LMDPE and the PLA, respectively.

$T_{g}$ value of PLA remains practically constant (60 °C) and is independent of the LMDPE addition and completely absent in the LMDPE curve, because the temperature range analyzed is outside of the LMDPE $\;T_{g}$. The presence of the cold crystallization peak ($T_{cc}$) and the cold crystallization enthalpy ($\Delta H_{cc}$) presented minimal signal in most of the samples, and only the ones which could be integrated are reported in Table 4. In the case of the melting peaks, blends of materials behave as physical mixtures. It was possible to measure independent peaks for the melting of PLA and LMDPE, where independent melting peaks for all the prepared mixtures of MB were integrated, with only minor changes for the values of $T_{m}$. Whereas for DB, an LMDPE melting peak for PLA75 blend could not be obtained.

The cold crystallization peak did not appear for some blends, several authors indicated that this is due to a very slow crystallization rate of PLA during cooling [24,38]. The crystallinity was calculated with the values of Table 4 and using Equation (1), neat PLA show low values, and these values increase for the blends as reported for other authors [16,33,39]. For example, Balakrishnan et al. [38] in PLA/LLDPE blends indicated that the LLDPE particles facilitate the mobility and rearrangement of PLA chains, resulting in increased crystallinity. The crystallinity also increases when the PLA and its blends are subjected to an additional thermal treatment, as is the case of melt blending.

### 3.4. Morphology

Figure 4 shows images of the rotomolded PLA prepared by DB (Figure 4a) and MB (Figure 4b). It can be seen that the parts made with PLA prepared by MB presents a yellowish coloration due to the thermal degradation of the polymer. During extrusion, mechanical, thermal, and oxidative degradation are important increases which are amplified by the subsequent rotational molding step. It has been proposed that thermal degradation mainly occurs by random main-chain scissions [30]. However, several reactions can occur such as hydrolysis, depolymerization, and oxidative degradation; as well as inter- and intramolecular reactions, which can all occur during PLA processing [29,30].

Optical micrographs of the internal surface of the rotomolded parts prepared by DB are shown in Figure 5 (left column). In these samples, particles of the minor phase have a spherical shape and present a broad size distribution with an average around 500 µm (initial particle sizes). It has been widely reported that this is a typical morphology of immiscible polymer blends [5,33,40,41], where large domains are obtained due to polymer incompatibility and lack of shear during mixing.

On the other hand, SEM micrographs were taken to observe the state of dispersion/adhesion of the phases in the blends prepared by MB. Figure 5 (right column) shows the SEM micrographs of cryo-fractured parts. It is clear that MB generates smaller particle sizes of the dispersed phase (for example PLA75 have a particle size of 16.2 ± 9.8 and PLA25 of 7.1 ± 4.3 µm) due to the higher stresses involved in the extruder. In all cases (DB and MB), a weak interfacial adhesion was produced between the PLA and LMDPE since no compatibilizer was used.

### 3.5. Thermal Degradation and Stability

The decomposition behavior and thermal stability of the different blends were studied by TGA under air atmosphere. Figure 6 shows the TGA and the derivative (DTG) curves for the neat polymers (Figure 6a,b) as well as for the blends (Figure 6c,d). To obtain more information about the thermal stability of these materials, three characteristic temperatures were extracted from these curves: (1) $T_{5}$ and (2) $T_{10},$ which are temperatures associated to a 5 and 10% weight loss, respectively, and (3) $T_{max},$ which is the temperature of maximum weight loss rate taken as the DTG peak [26,33,37]. The results are reported in Table 5, where the neat polymers exhibited a slight difference in degradation behavior under both mixing methods (Figure 6a,b). PLA prepared by MB (PLA_MB) starts to decompose at 277 °C, with a maximum weight loss rate at 335 °C; while PLA prepared by DB (PLA_DB) starts to decompose at 286 °C and also presents a single step degradation process with a maximum weight loss rate at 338 °C. On the other hand, LMDPE shows multiple steps degradation process with a $T_{max}$ of 358 °C for both samples (LMDPE_DB and LMDPE_MB). PLA/LMDPE blends also exhibited a multiple step degradation process (Figure 6c,d). The first step is related to PLA degradation, and its magnitude increases with the PLA content in the blend.

In general, the characteristic temperatures shifted to a lower value for the materials prepared by MB as a result of a higher degradation due to their processing. Cuadri and Martín-Alfonso reported that a thermo-oxidative degradation occurs because of high temperatures and the presence of oxygen, with chain scission as the principal mechanism of degradation [31]. After both processing steps (extrusion and rotomolding), melt blended materials with higher PLA content experiment higher degradation level negatively affecting their appearance and performance.

Extrusion times are crucial in the processing of PLA, and longer times associated to induce a higher thermal degradation [42]. Lower residence times, which can be achieved by modifying the length and/or diameter of the extrusion screw (L/D), could decrease the degradation of PLA [43]. However, the use of an L/D of 40 guarantees a better dispersion and distribution of the dispersed phase [42,43].

### 3.6. Mechanical Properties

#### 3.6.1. Tensile Properties

Figure 7 shows the tensile modulus and tensile strength of the rotomolded parts prepared by DB and MB. Both values for the blends are between the pure components, which represents a combination of the PLA stiffness with the LMDPE toughness. The blends show a slight modulus increase for the parts prepared by MB. DB PLA showed an increase in tensile strength and tensile modulus of 54% (from 29.6 to 45.6 MPa) and 14.6% (from 1240 to 1421 MPa), respectively, as a result of a higher level of degradation presented in the MB PLA. A similar trend was observed for PLA87, which presented a 11% increase in the tensile strength (from 33.9 to 37.7 MPa), for the sample prepared by DB compared with the sample prepared via MB. Taubner and Shishoo [29] reported that PLA tensile strength is influenced by processing conditions during extrusion. PLA tensile strength value decreased 39% (from 58.5 to 35.6 MPa) for dry granules extruded at 240 °C, compared with granules extruded at 210 °C.

#### 3.6.2. Flexural Properties

Figure 8 shows the flexural modulus and strength of the rotomolded parts prepared by DB and MB. Similar to the tensile properties, the flexural strength of PLA and the blend PLA87 prepared by DB showed an increase of 111% (from 33.8 to 71.4 MPa) and 37.6% (from 39.9 to 54.9 MPa), respectively, compared to the samples prepared by MB. LMDPE does not present significant changes upon the preparation method. As expected, the flexural modulus and strength of the blends decreased with increasing the LMDPE content. This can be correlated with higher flexural properties of the neat PLA. Except for neat PLA and blend PLA87, slight differences in flexural modulus and strength were observed between both mixing strategies.

#### 3.6.3. Impact Properties

The Charpy impact strength of the rotomolded parts in Figure 9 shows a significant increase in this property. The values go from 28.5 J/m for neat PLA to 130.5 J/m for neat LMDPE, and the material is going from a rigid/fragile one to a tough/ductile one. An interesting synergy effect is observed in a certain composition in both preparation techniques (DB and MB). For example, at PLA12 prepared by MB, an increase of 24% (from 117.7 to 146 J/m) compared with neat LMDPE and 440% (from 27.2 to 146 J/m) compared with neat PLA. On the other hand, in the blend with PLA25 prepared by DB, the increase was up to 40% (from 130.5 to 182.2 J/m) compared with neat LMDPE and 550% (from 28.5 to 182.2 J/m) compared with neat PLA. Anderson and Hillmyer [21] prepared PLA/LLDPE blends by internal mixing chamber followed by compression molding and reported that the addition of 20 wt. % of LLDPE to a PLA matrix results in an increase of 1650% (from 20 to 350 J/m) in Izod impact strength. In a similar way, Balakrishnan et al. [38] prepared PLA/LLDPE blends by injection molding and correlated the Izod impact strength and LLDPE particle size dispersed in PLA matrix. They reported an increase of 53% (at 10 wt. % LLDPE) attributed to LLDPE smaller particle size dispersed in the PLA matrix.

## 4. Conclusions

In this work, blends of PLA/LMDPE were processed by rotational molding. Prior to the rotomolding process, the blends were prepared by a simple dry-blending (DB) or an extrusion based melt-blending (MB) process to determine the effect of the mixing strategy on the final thermal and mechanical properties of the rotomolded samples. The rheological properties of neat materials processed, either via MB or DB, showed superimposed complex viscosity values for all the temperatures tested, except for 220 °C, where the complex viscosity of the MB preparation decreased. For blends, the complex viscosity at 200 °C showed that viscosity values were higher for PLA25 and PLA50 via MB, viscosity values compared to DB samples. In contrast, viscosity values for PLA75 were higher for DB than MB.

The MB strategy produced particles with a better dispersion and smaller sizes, where values <1 mm could be observed in SEM micrographs, in contrast to DB fabricated pieces, which could be observed with an optical microscope (>1 mm particles). However, MB fabricated pieces showed clear signs of thermal degradation, whereas DB fabricated were clearer and translucent. The thermal characteristics measured using DSC were similar for both strategies, with minor changes in the measured peaks and crystallinity values. In contrast, TGA studies of MB samples showed a minor significant shift at lower degradation temperatures.

The mechanical properties of the rotomolded parts were influenced by the mixing technique and PLA content. The tensile and flexural modulus showed a slight increase for MB preparations in almost all the characterized blends. For MB and DB fabricated samples, the highest tensile and flexural strength values were observed for neat DB PLA, followed by the DB PLA87 blend. Finally, a synergic effect for the Charpy impact strength was observed for the blend with PLA25 prepared by DB and blend with PLA12 prepared by MB, where the increase was up to 550% and 440%, respectively, compared with neat PLA.

The dry blending approach represents a faster and easier fabrication method to produce blends for rotomolding technology, including cases where one of the components has a low thermal stability, as seen for PLA. With this methodology, the material degradation is avoided, and, for certain specific blends, the mechanical properties are improved.

## Figures and Tables

**Figure 1 polymers-13-00217-f001:**
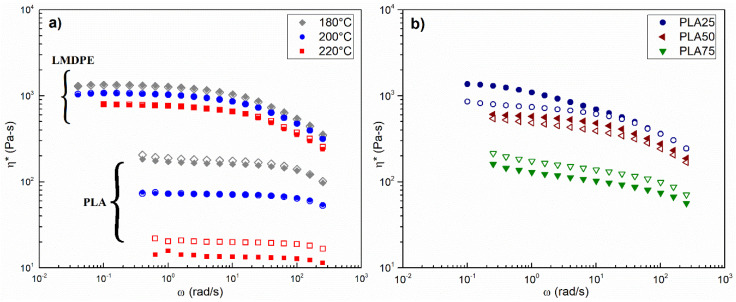
Complex viscosity as a function of frequency for (**a**) neat PLA and LMDPE at different temperatures, (**b**) blends at 200 °C. Open symbols are for dry-blending (DB) and closed symbols for melt-blending (MB) mixing strategy. In neat materials, DB and MB are completely superimposed at several frequency values.

**Figure 2 polymers-13-00217-f002:**
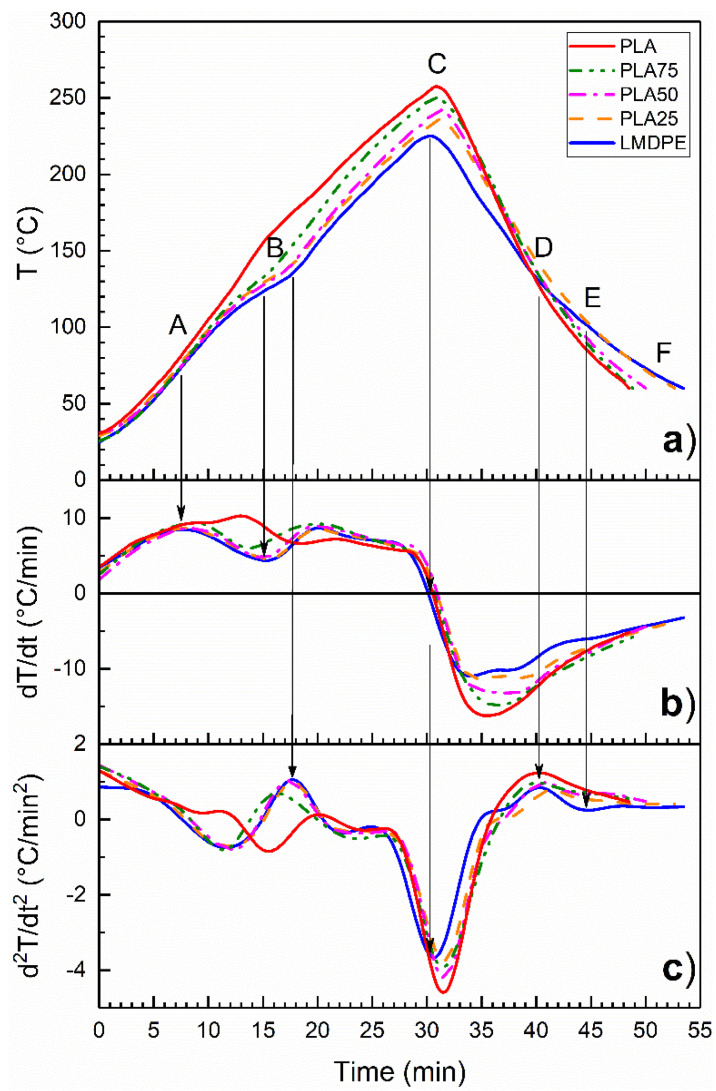
IAT (internal air temperature) profiles (**a**), as well as their first (**b**) and second (**c**) derivatives as a function of time for PLA, LMDPE, and their blends prepared by dry blending (DB).

**Figure 3 polymers-13-00217-f003:**
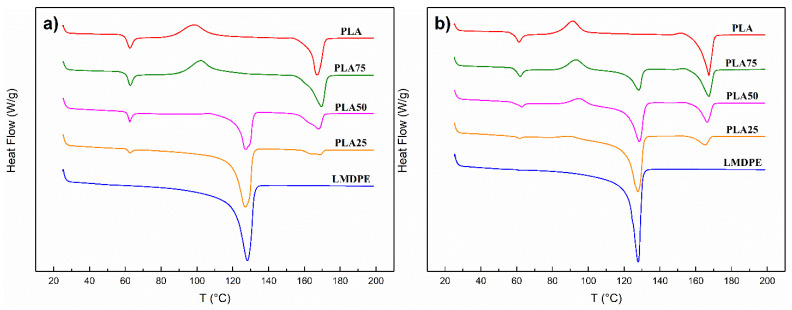
Typical differential scanning calorimetry (DSC) curves of PLA/LMDPE rotomolded blends prepared by (**a**) DB and (**b**) MB.

**Figure 4 polymers-13-00217-f004:**
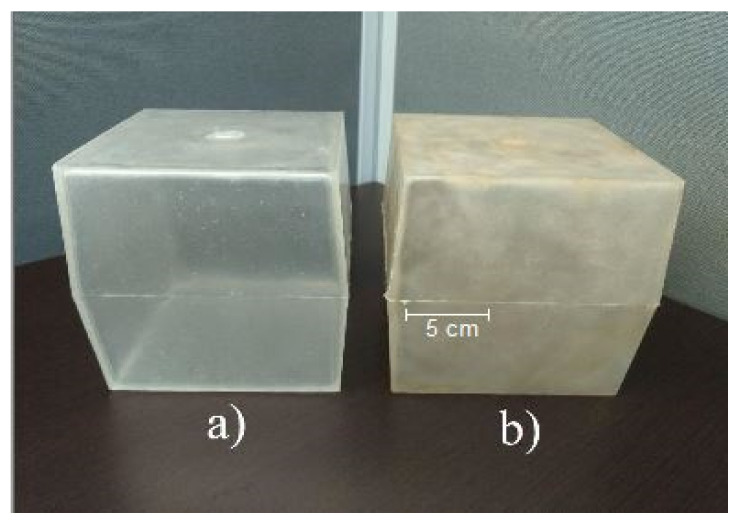
Rotomolded PLA pieces prepared by (**a**) DB and (**b**) MB.

**Figure 5 polymers-13-00217-f005:**
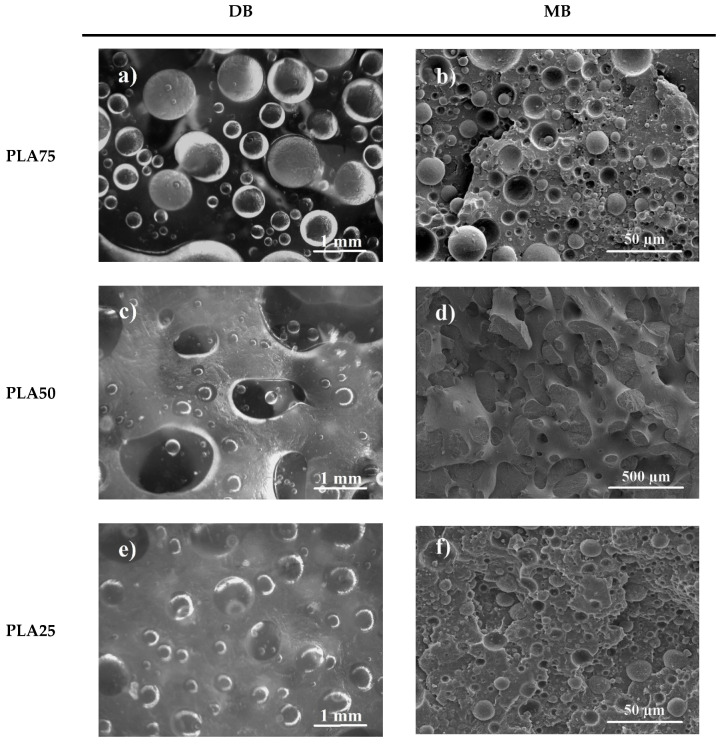
Optical images of the internal surface of rotomolded parts prepared by DB (**left column**) and SEM micrographs of MB cryo-fractured parts (**right column**): (**a**,**b**) PLA75, (**c**,**d**) PLA50, and (**e**,**f**) PLA25.

**Figure 6 polymers-13-00217-f006:**
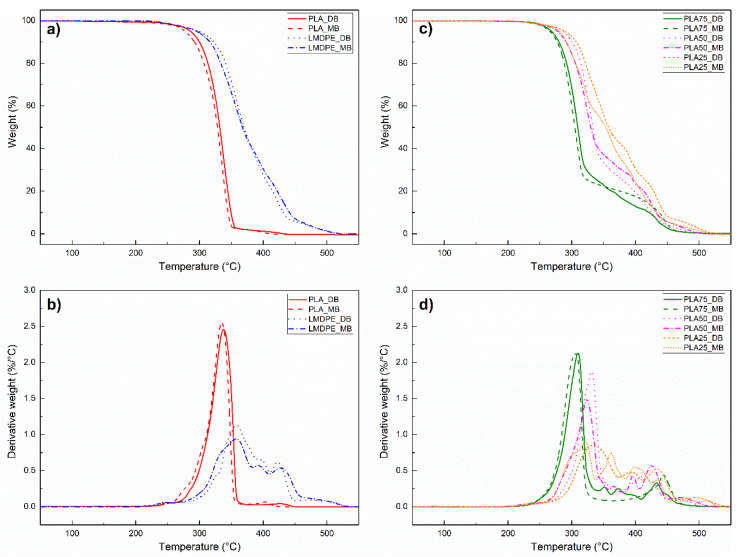
TGA (thermogravimetric analysis) curves for (**a**) neat polymers, (**c**) polymer blends and the derivative (DTG) curves for (**b**) neat polymers, (**d**) polymer blends prepared by dry (DB) and melt blending (MB).

**Figure 7 polymers-13-00217-f007:**
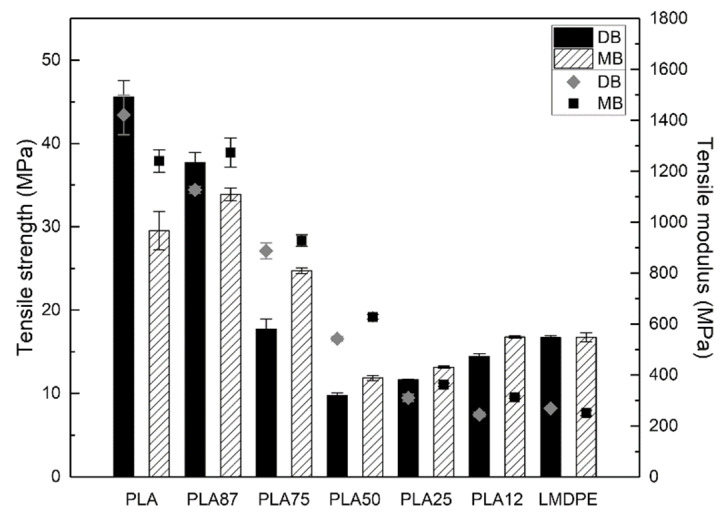
Tensile properties of the rotomolded parts: Strength (column bars) and modulus (closed symbols).

**Figure 8 polymers-13-00217-f008:**
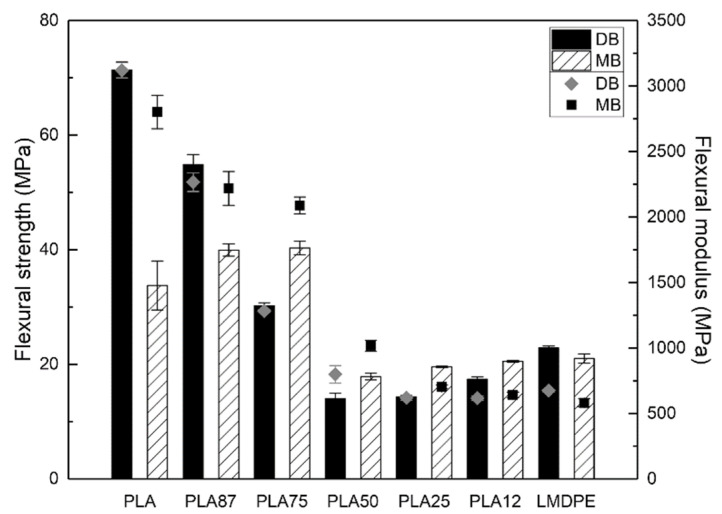
Flexural properties of the rotomolded parts: Strength (column bars) and modulus (closed symbols).

**Figure 9 polymers-13-00217-f009:**
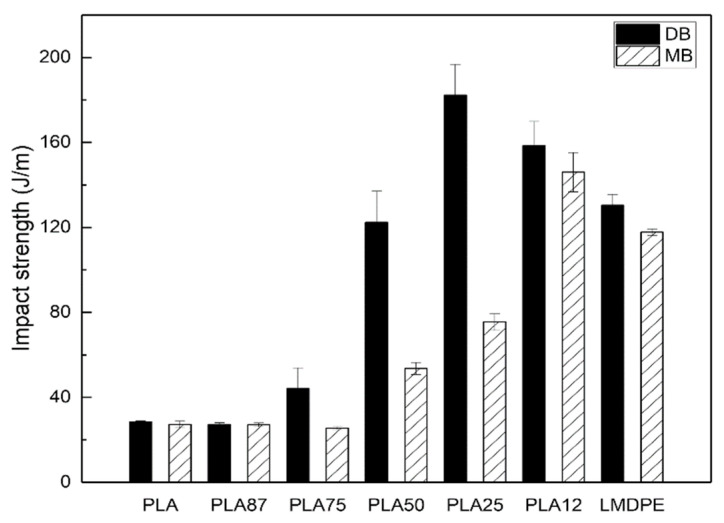
Impact strength of the rotomolded parts.

**Table 1 polymers-13-00217-t001:** Particle size distribution of the polymers. PLA: Poly(lactic acid); LMDPE: Linear medium density polyethylene.

Particle Size (µm)	850–425	425–300	300–212	212–150	150–75
PLA (%)	74.6	25.4	—	—	—
LMDPE (%)	12.4	40.4	25.2	16.9	5.1

**Table 2 polymers-13-00217-t002:** Composition of the PLA/LMDPE blends.

Sample	PLA	PLA75	PLA50	PLA25	LMDPE
PLA (wt. %)	100	75	50	25	0
LMDPE (wt. %)	0	25	50	75	100

**Table 3 polymers-13-00217-t003:** Experimental measured density and average thickness of the rotomolded parts.

Sample	$\mathrm{Density}\;\left(\frac{g}{cm^{3}}\right)$		Average Thickness (mm)	
	DB	MB	DB	MB
PLA	1.23 ± 0.04	1.24 ± 0.03	2.48 ± 0.41	2.42 ± 0.61
PLA75	1.20 ± 0.05	1.11 ± 0.01	2.31 ± 0.17	2.54 ± 0.07
PLA50	1.04 ± 0.02	1.03 ± 0.01	2.64 ± 0.15	2.68 ± 0.20
PLA25	0.98 ± 0.02	0.96 ± 0.01	2.89 ± 0.12	3.02 ± 0.12
LMDPE	0.93 ± 0.01	0.92 ± 0.01	3.08 ± 0.01	3.27 ± 0.18

**Table 4 polymers-13-00217-t004:** Thermal properties of the rotomolded PLA, LMDPE, and their blends.

Sample	$T_{g}\;({}^{\circ}C){\;}^{a}$	Cold Crystallization ^a^		Melting		Crystallinity (%)
		$T_{cc}\;({}^{\circ}C)$	$\Delta H_{cc}\;(J/g)$	$T_{m}\;({}^{\circ}C)$	$\Delta H_{m}\;(J/g)$	
DB						
PLA	59.4	98.9	33.4	166.9	41.8	8.9
PLA75	60.2	102.4	36.8	169.5 ^a^	42.2 ^a^	7.7 ^a^
PLA50	60.8	—	—	168.0 ^a^/127.7 ^b^	21.9 ^a^/51.9 ^b^	46.7 ^a^/36.0 ^b^
PLA25	60.6	—	—	169.0 ^a^/126.9 ^b^	7.8 ^a^/99.8 ^b^	33.3 ^a^/46.2 ^b^
LMDPE	—	—	—	128.2	130.8	45.4 ^a^
MB						
PLA	58.5	91.3	29.0	167.4	44.2	16.2
PLA75	58.8	93.2	20.3	167.4 ^a^/128.1 ^b^	32.0 ^a^/26.8 ^b^	16.6 ^a^/37.2 ^b^
PLA50	58.2	95.3	10.9	166.4 ^a^/128.5 ^b^	21.3 ^a^/55.2 ^b^	22.2 ^a^/38.3 ^b^
PLA25	58.8	—	—	165.2 ^a^/127.6 ^b^	10.4 ^a^/104.5 ^b^	44.4 ^a^/48.4 ^b^
LMDPE	—	—	—	127.9	153.2	53.2

^a^ PLA, ^b^ LMDPE.

**Table 5 polymers-13-00217-t005:** TGA data of neat polymers and PLA/LMDPE blends.

Sample	$T_{5}$ (°C)		$T_{10}$ (°C)		$T_{max}$ (°C)	
	DB	MB	DB	MB	DB	MB
PLA	286	277	300	292	338	335
PLA75	268	266	280	277	310	306
LA50	286	278	300	291	331	324
PLA25	289	280	307	292	333	321
LMDPE	299	296	322	316	358	358

## Data Availability

The data presented in this study are available on request from the corresponding author.

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
