# Peer review of "Rotational Molding of Poly(Lactic Acid)/Polyethylene Blends: Effects of the Mixing Strategy on the Physical and Mechanical Properties"

_polymers, 2021, doi:10.3390/polym13020217_

Round 1
Reviewer 1 Report
In this work by Ruiz-Silva and colleagues, the authors studied the processing of blends of poly(lactic acid)/polyethylene by rotational molding. They investigated how dry-blending or melt-blending as mixing strategies impact various properties of the material. The presented reasoning is sound and well supported with evidence. I would be happy to recommend this paper for publication in Polymers should the authors incorporate the following minor suggestions:
1) It would be beneficial to provide at least one sentence long description of this contribution's novelty factor in the "in this work" section of the introduction. This will enable readers to quickly judge if this article is relevant and interesting to read.
2) What was the nitrogen flow rate in TGA experiments? What was the purity of nitrogen used for experiments? (2.5.1.)
3) Similarly, what was the flow rate of air? (2.5.2.) The decomposition profile is often dependent on the amount of air supplied to the system.
4) Fig. 1 is blurred. Please ensure that the plots are sharp in the revised version of the manuscript.
5) Plots should have uniform size and formatting to make the article easy to follow. It is essential for contributions like this, which contains many plots and a thorough description of the material's characterization from various angles.
Author Response
The answers are in the attached document, thank you for your valuable comments.

Reviewer 2 Report
Some data are missing for the clear presentation of the entire researched topic. This needs to be improved and/or clarified. Remained comments are in the attachment.

Author Response

(The authors gave the same response as above.)

Reviewer 3 Report
This paper is a very valuable comparative analysis of two types of material preparation strategies in the rotomolding technique. Due to the small amount of research relating to processing by rotomolding, this work is a valuable contribution to the development of methods of materials processing used in this technique. The presented methodology of work is very clear, the description and presentation of the results are also at a high level.
I think, however, that the presentation of the results of micro and macroscopic observations of samples should be supplemented with some more pictures.
The use of polymer mixtures should significantly limit transparency, therefore the view of samples could be valuable.
SEM pictures compilation should also presents dry-blend samples.
Taking into account the visible porosity, it would also be helpful to indicate the differences in density between DB and MB samples.
Author Response

(The authors gave the same response as above.)

Round 2
Reviewer 2 Report
Dear authors,
you have clarified the majority of the open questions. There are still some minor tasks to solve:
- volume is measured in ml, not mL - correct this in 172 and 185
- Equation 3: you present double exponent - is this correct so?
- the selected reference nr. 42 could be also applicable but use a significantly larger L/D ratio as the proposed reference doi:10.3390/polym11081248. I suggest you use both of them. In the reference list, the stated reference 42 has the wrong doi number.
Author Response
Dear authors,
you have clarified the majority of the open questions. There are still some minor tasks to solve:
1. volume is measured in ml, not mL - correct this in 172 and 185
Done
2. Equation 3: you present double exponent - is this correct so?
Corrected, thank you!
3.the selected reference nr. 42 could be also applicable but use a significantly larger L/D ratio as the proposed reference doi:10.3390/polym11081248. I suggest you use both of them. In the reference list, the stated reference 42 has the wrong doi number.
The doi was corrected and included the proposed reference.